# The Effect of Neuropsychiatric Drugs on the Oxidation-Reduction Balance in Therapy

**DOI:** 10.3390/ijms25137304

**Published:** 2024-07-03

**Authors:** Karina Sommerfeld-Klatta, Wiktoria Jiers, Szymon Rzepczyk, Filip Nowicki, Magdalena Łukasik-Głębocka, Paweł Świderski, Barbara Zielińska-Psuja, Zbigniew Żaba, Czesław Żaba

**Affiliations:** 1Department of Toxicology, Poznań University of Medical Sciences, 3 Rokietnicka Street, 60-806 Poznań, Poland; 2Department of Forensic Medicine, Poznań University of Medical Sciences, 10 Rokietnicka Street, 60-806 Poznań, Poland; 3Department of Emergency Medicine, Poznań University of Medical Sciences, 7 Rokietnicka Street, 60-806 Poznań, Poland

**Keywords:** neuropsychiatric drugs, oxidative stress, first- and second-generation antipsychotics, mood stabilizers, antidepressants

## Abstract

The effectiveness of available neuropsychiatric drugs in the era of an increasing number of patients is not sufficient, and the complexity of neuropsychiatric disease entities that are difficult to diagnose and therapeutically is increasing. Also, discoveries about the pathophysiology of neuropsychiatric diseases are promising, including those initiating a new round of innovations in the role of oxidative stress in the etiology of neuropsychiatric diseases. Oxidative stress is highly related to mental disorders, in the treatment of which the most frequently used are first- and second-generation antipsychotics, mood stabilizers, and antidepressants. Literature reports on the effect of neuropsychiatric drugs on oxidative stress are divergent. They are starting with those proving their protective effect and ending with those confirming disturbances in the oxidation–reduction balance. The presented publication reviews the state of knowledge on the role of oxidative stress in the most frequently used therapies for neuropsychiatric diseases using first- and second-generation antipsychotic drugs, i.e., haloperidol, clozapine, risperidone, olanzapine, quetiapine, or aripiprazole, mood stabilizers: lithium, carbamazepine, valproic acid, oxcarbazepine, and antidepressants: citalopram, sertraline, and venlafaxine, along with a brief pharmacological characteristic, preclinical and clinical studies effects.

## 1. Introduction

Mental disorders have constituted a burden on society for many years. The situation has deteriorated significantly as a result of the pandemic, as well as the risks associated with the increasing stress and direct neuropsychiatric effects of the SARS-CoV-2 virus. Unfortunately, the efficacy of available neuropsychiatric drugs is not sufficient in an era of increasing numbers of patients, and the involvement of both diagnostically and therapeutically complex diseases is rising. Psychiatric disorders are a range of various conditions associated with abnormal functioning of the nervous system, which are common in patients with nervous system disorders [1]. As a result of these abnormalities, in response to a stressful situation, the body activates maladaptive behavioral patterns that impede daily functioning [2]. The most common mental disorders include mood swings, neurotic disorders, dementia disorders, schizoaffective disorders, eating and personality disorders, sleep disorders, or addictions (Table 1) [3,4,5,6,7,8,9,10,11,12,13,14,15,16,17,18,19,20,21,22,23]. According to the World Health Report, mental disorders affect approximately 25% of people at some point during their lifetime. The co-occurrence of neurological diseases with mood disorders and anxiety syndromes is common in patients with neurodegenerative diseases (Parkinson’s, Alzheimer’s, and Huntington’s), stroke, epilepsy, and demyelinating diseases. The choice of treatment for neurological patients with co-occurring mood and anxiety disorders is often difficult, as it has to take into account the symptoms of the underlying disease and poorer drug tolerance [1]. According to the International Neuropsychiatry Association, discoveries in the pathophysiology of neuropsychiatric diseases are promising, including those initiating a new round of innovation within the contribution of oxidative stress to the etiology of neuropsychiatric diseases [2,24,25]. Oxidative stress is a significant risk factor for many psychiatric disorders, for the treatment of which first- and second-generation antipsychotics, mood stabilizers, and antidepressants are most commonly used (Table 2) [2]. 

Oxidative stress is a state of imbalance between antioxidants and free radicals in the body. Both oxidants and antioxidants, essential for vital processes, should remain in quantitative balance for the safety of the entire system. However, there are known factors under the influence of which one overpowers the other, resulting in changes, i.e., DNA damage, disruption of protein, lipid, or carbohydrate function and structure. In addition, oxidative stress may be involved in the etiology of many conditions, i.e., hypertension, obesity, atherosclerosis, or diabetes. Also, in terms of many neuropsychiatric diseases, despite advances in neurobiological research, the pathophysiology of many of these diseases has yet to be more extensively described. Advances in the understanding of the pathomechanism of many psychiatric disorders, as well as neuropsychiatric diseases, may help to discover effective therapies and indicators for their early diagnosis. Oxidative stress, which damages biomolecules and causes dysfunctions of, among others, cellular systems at the level of, e.g., mitochondria or dopamine receptors, is also referred to as a critical phenomenon in patients with mental disorders [2,26,27,28,29] (Figure 1). 

Under conditions of homeostasis, and thus at safe concentrations, the presence of reactive oxygen species (ROS) is not harmful to the organism and acts as a regulator and mediator of physiological processes. ROS play an essential role in the ordinary course of inflammatory reactions, regulation of immune processes, or T-lymphocyte activation, with concomitant adhesion of leukocyte cells to the endothelium [27,28,29,30]. Currently, the diagnostic market makes available numerous analytical tools and biomarkers that can be used to assess oxidative stress. Examples of oxidative markers include myeloperoxidase (MPO), 4-hydroxynonenal (HNE), F2-isoprostanes (IsoP), 8-hydroxy-2-deoxyguanosine (8-OHdG), malondialdehyde (MDA), allantoin, and thiobarbituric acid reactive substances (TBARS). They are used to monitor tissue damage resulting from a loss of balance between antioxidants and free radicals [31,32,33]. Table 3 brings together the oxidative stress markers according to the adverse changes they are used to assess. 

Knowledge of the biotransformation of drugs used to treat psychiatric disorders involving enzymatic systems may support the hypothesis of oxidative disorders generated by, among other things, reactive metabolites or radicals. However, the link between the effects of many drugs and oxidative stress is still the subject of few scientific publications aiming to confirm their antioxidant or pro-oxidant effects in groups of treated patients [35,36]. 

The presented publication reviews the state of knowledge on the share of oxidative stress in the most commonly used therapies for neuropsychiatric diseases using first- and second-generation antipsychotics, i.e., haloperidol (HLP), clozapine (CLO), risperidone (RIS), olanzapine (OLZ), quetiapine (QT) or aripiprazole (ARP), mood stabilizers: lithium (Li), carbamazepine (CBZ), valproic acid (VPA), oxcarbazepine (OXC), and antidepressants: citalopram (CIT), sertraline (SER), venlafaxine (VEN), along with an explanation of the obtained results and their comparison. The data provided and discussed come from the US National Library of Medicine (PubMed) bibliographic sources, selecting publications from the last ten years and older for analysis in order to discuss the results of more recent preclinical and clinical studies. The key search terms for the literature search were oxidative stress and neuropsychiatric drugs.

## 2. First and Second-Generations of Antipsychotic Drugs

### 2.1. Haloperidol (HLP)

#### 2.1.1. Pharmacological HLP Data

Haloperidol (HLP), synthesized in 1958, belongs to the butyrophenone derivatives. Its discovery came during research into new derivatives of pethidine and methadone, both of which were claimed to have analgesic activity. However, it has been proven that this drug has significant efficacy in the treatment of psychiatric disorders, including schizophrenia, especially hallucinations and delusions [37,38,39]. The antipsychotic effect of the drug is due to the blockade of dopamine D_2_ receptors. Its high affinity for the receptor, together with a low cholinolytic effect, minimizes the occurrence of symptoms such as constipation, dry mouth, and visual disturbances [38,40,41,42,43]. 

#### 2.1.2. Preclinical HLP Studies

Oxidative stress occurring with a first-generation antipsychotic drug such as haloperidol is not a fully understood issue. The use of the drug has been shown to increase oxidative stress, which has an impact on the pathophysiology of schizophrenia. On the one hand, oxidative imbalance may be part of the pathogenesis of the disease, but on the other hand, it can lead to patients experiencing extrapyramidal symptoms (EPS). Preclinical and in vitro (cell) studies conducted on HLP have revealed that the use of the drug is associated with a decrease in antioxidant enzyme activity and an increase in reactive oxygen species (ROS) levels [44,45,46,47,48,49,50,51]. Perera et al. demonstrated that biochemical changes in the rat brain after haloperidol administration are accompanied by histopathological changes such as necrosis and fibrosis. Although the exact pathophysiology of EPS is not known, the authors here point to an important role for ROS in inducing oxidative stress, representing one mechanism of haloperidol-induced long-term neurotoxicity [44]. When HLP is used, dopamine is increased, which deaminates and generates significant amounts of free radicals and hydrogen peroxide, among other things. Reactive metabolites leading to multicellular damage and lipid peroxidation contribute to the cytotoxic effects of the drug. Raudenska et al. showed that long-term haloperidol administration to rats leads to changes in sigma-1R receptor and inositol triphosphate receptor (IP3R) expression, modifying myocardial contractility. This may underlie cardiac arrhythmias and QT interval prolongation in patients using haloperidol [45]. Haloperidol can also lead to changes in brain tissue, as evidenced by the study by El-Awadan et al. After 14 days of administering the drug to rats at a dose of 1 mg/kg, the concentrations of GSH, malondialdehyde (MDA), and nitric oxide (NO) in the brain and liver were determined. Levels of the former were significantly reduced in both tissues. In contrast, there was an increase in NO and MDA levels in brain tissue [47]. The adverse effects of the drug on antioxidant enzymes in the brain were also confirmed by measuring the expression of the enzyme peroxiredoxin-6. After a 28-day administration of haloperidol (1 mg/kg) to rats, significantly increased levels of the test protein were observed in the grey matter of the brain, with a concomitant increase in lipid peroxidation. However, peroxiredoxin-6 concentrations were unchanged in the white matter of the brain and liver of rats [48]. 

Subsequent studies in human cell lines confirmed that, under prooxidant conditions, therapeutic concentrations of haloperidol increase TBARS levels. However, a significant correlation between the used drug and lipid peroxidation was not observed for cells not treated with hydrogen peroxide. In addition, the combination of lithium and haloperidol significantly reduced cell survival. Despite the lack of statistical significance of the results obtained, they indicate the need for particular caution when combining drugs used in the treatment of psychiatric disorders [50,51].

#### 2.1.3. Clinial HLP Studies

Available clinical data on the relationship between HLP and oxidative stress points to the use of biomarkers such as superoxide dismutase (SOD), catalase (CAT), and glutathione (GSH). The use of haloperidol results in a decrease in the activity of the above antioxidant enzymes, as confirmed by a study by Bošković et al. conducted on a group of patients treated with haloperidol for a minimum of six months. In addition, they observed significantly lower concentrations of the biomarkers determined in patients with extrapyramidal symptoms [46]. In contrast, Singh et al. further assessed markers of lipid peroxidation by determining serum thiobarbituric acid reactive substances (TBARS). TBARS levels were significantly elevated in patients treated with haloperidol (for at least six weeks) compared with patients taking the second-generation antipsychotic drug olanzapine. SOD has also been shown to be negatively correlated with serum TBARS [49]. 

### 2.2. Clozapine (CLO)

#### 2.2.1. Pharmacological CLO Data

Clozapine (CLO), which belongs to the atypical antipsychotics, has structural similarities to tricyclic antidepressants. Its synthesis took place in the early 1960s. A clear advantage of clozapine appeared to be the absence of extrapyramidal side effects. Nonetheless, the drug was initially skeptically assessed and withdrawn from use, a factor contributed to by the significant risk of agranulocytosis during its use. However, continued research has confirmed significant effectiveness in the treatment of symptoms of psychosis, especially drug-resistant schizophrenia [52,53]. Regular blood tests of patients have also been shown to control the risk of agranulocytosis. Consequently, the drug was reintroduced to the pharmaceutical market in 1989 in Europe—and a year later in the USA. However, its use is limited, and it is not a first-line drug. The inclusion of CLO in pharmacotherapy occurs primarily when treatment with other antipsychotics is unsuccessful. In addition, its use is associated with adverse effects that are troublesome for patients, e.g., weight gain, constipation, metabolic disorders, and the development of diabetes [52,54,55]. The antipsychotic effect of clozapine is due to its diverse pharmacological profile. This is because it shows affinity for dopamine D_1_, D_4_, and D_2_ receptors. In the case of the D_2_ receptor, dissociation occurs very rapidly, thus reducing the risk of hyperprolactinemia and extrapyramidal symptoms. Clozapine also has an affinity for serotonin, adrenergic, histamine, and muscarinic receptors [52,54,55].

#### 2.2.2. Preclinical CLO Studies

There are numerous attempts to link oxidative stress to adverse effects, including metabolic disorders occurring during clozapine use, in preclinical studies. This points to the involvement of free radicals leading to the oxidation of proteins, which results in their disruption or loss of function [56,57,58,59,60,61]. In the study by Walss-Bass et al., human neuroblastoma cells were used to identify changes in proteins under oxidative stress after administration of CLO (in three concentrations of 1, 10, or 20 µM). The level of reactive oxygen species increased with dose until reaching a maximum value at a drug concentration of 10 µM. Among others, malate dehydrogenase, calumenin, or the muscle isoenzyme of keratin kinase were found to be particularly susceptible to oxidation. Irreversible changes in enzymes involved in key metabolic processes can lead to the occurrence of multiple disorders in the body [57]. 

A study by Abdel-Wahab et al. in rats further revealed an association between oxidative stress and myocardial ischemia. Clozapine causes an increased release of catecholamines, which in turn increases the production of free radicals. As a consequence, cell damage occurs with the infiltration of neutrophils, which release cytokines, among others, TNF-α. Rats treated with CLO for 21 days showed a significant increase in MDA and 8-hydroxy-2-deoxyguanosine (8-OHdG), which was observed at doses of 15 and 25 mg/kg/day. This indicates lipid peroxidation and oxidative damage to DNA. The cardiotoxic effect was further confirmed by an increase in the activity of the biomarkers of cardiac damage—lactate dehydrogenase (LDH) and cardiac creatine kinase isoenzyme (CK-MB). At the same time, GSH levels decreased significantly [58]. Similar results were obtained during clozapine cardiotoxicity studies using striped danios. After a three-day exposure of embryos to the drug (concentration range 12.5–100 µM), cardiac morphological abnormalities and bradycardia were detected. In addition, levels of ROS, MDA, and pro-inflammatory cytokines increased. In contrast, CAT and SOD activity decreased. These results confirmed that CLO contributes to both oxidative stress and inflammation, thus leading to cardiotoxicity in preclinical studies [59]. 

#### 2.2.3. Clinial CLO Studies

Clozapine-induced oxidative stress also affects blood cells in patients with agranulocytosis and in patients without agranulocytosis. A clinical study by Fehsel et al. suggests that the use of the drug is associated with an increase in ROS production in neutrophils, which may contribute to the induction of their apoptosis. In addition, increased expression of the proapoptotic gene p53 was observed, and this also occurred in patients treated with other antipsychotics, including olanzapine [60]. 

However, the results of a study comparing oxidative stress parameters in patients treated with perphenazine, risperidone, and clozapine led the authors to a completely different conclusion. Indeed, it has been shown that in patients treated with clozapine for at least three months, serum GSH and SOD levels increased, and MDA levels were significantly lower compared to patients treated with perphenazine and risperidone [61]. 

### 2.3. Risperidone (RIS)

#### 2.3.1. Pharmacological RIS Data

During studies conducted in the 1960s and 1970s, it was observed that the administration of lysergic acid diethylamide (LSD) to rats produced symptoms similar to the negative symptoms of schizophrenia. This phenomenon has been linked to 5-HT_2_ receptor agonism and serotonin overactivation. A new target in the field of treatment of psychotic disorders has, therefore, become the development of compounds that block D_2_ and 5-HT_2_ receptors simultaneously. In the early 1980s, a benzoxazole derivative was obtained, which was classified as an atypical antipsychotic drug in the form of risperidone (RIS). The drug was approved by the US Food and Drug Administration in a relatively short time, as early as 1993, for the treatment of schizophrenia [62,63,64]. The drug also binds to α_1_adrenergic receptors and α_2_ and histamine H_1_ receptors. A particular advantage of the drug is its lack of affinity for muscarinic receptors; thus, there is little likelihood of anticholinergic symptoms developing. Although it is not without side effects and can lead to weight gain, hyperprolactinemia, hyperlipidemia, and diabetes, among others, it has a limited side effect profile compared to its predecessors [64,65,66,67,68]. 

#### 2.3.2. Preclinical RIS Studies

With risperidone, the incidence of serious side effects such as extrapyramidal symptoms is limited, which is attributed to lower levels of oxidative stress [69,70,71,72]. Stojkovic et al. demonstrated that the drug affects parameters such as MDA, GSH, and SOD in the brain structures of rats. An increase in previously reduced GSH levels was observed in the brains of rats administered phencyclidine (PCP) initially for redox imbalance, followed by RIS for nine weeks. PCP led to oxidative damage in the brain and lipid peroxidation, but long-term administration of risperidone resulted in reduced levels of MDA and SOD [69]. 

However, some researchers indicate that risperidone has a cytotoxic effect by increasing the production of reactive oxygen species, thus inducing oxidative stress. In isolated hepatocytes of lab rats incubated with risperidone, a significant decrease in GSH levels was observed compared to the control group, as well as signs of cell damage, including, among others, a decrease in mitochondrial membrane potential and lipid peroxidation [72].

#### 2.3.3. Clinial CLO Studies

Similar results were observed in a group of patients treated with RIS for 12 weeks at a dose of 6 mg/day. The concentration of SOD in the blood decreased significantly. However, the effect of the drug on NO levels remains inconclusive. The above study did not show a decrease in elevated NO levels in patients’ blood [70]. Different results were obtained in the blood of patients undergoing six weeks of pharmacotherapy with risperidone at individually selected doses in the range of 3–11 mg/day, as there was an increase in the levels of NO and its metabolites. At the same time, NO levels in the blood of patients with schizophrenia have also been shown to be significantly lower than in healthy individuals [71]. The aforementioned study by Hendouei et al. also assessed the effect of risperidone on oxidative stress parameters. Although clozapine showed a significant advantage in antioxidant activity, the total antioxidant capacity of plasma (FRAP, ferric reducing antioxidant power) reached the highest value compared to the patient group treated with risperidone [61]. 

### 2.4. Olanzapine (OLZ)

#### 2.4.1. Pharmacological OLZ Data

In 1996, olanzapine (OLZ) was approved by the FDA for the treatment of schizophrenia, and in 2004, it was also approved for the treatment of bipolar disorder [73,74]. Due to its structural similarity to clozapine, olanzapine has a similar receptor profile. It shows high affinity for both dopamine receptors (e.g., D_2_, D_4_), serotonin (among others, 5-HT_2A_, 5H-T_2C_), adrenergic, histamine, as well as muscarinic. It has been shown that this drug also affects the glutamatergic system by interacting with NMDA receptors. However, the exact mechanism is not known [74,75]. The use of OLZ is associated with a limited risk of extrapyramidal symptoms and small increases in prolactin levels. Despite the above-mentioned advantages, the drug, especially during long-term therapy, significantly interferes with the body’s metabolic processes, leading to numerous disorders such as hyperlipidemia, diabetes, and weight gain. The latter is more common with olanzapine compared to other antipsychotics. Drowsiness, dry mouth, dizziness, and fatigue, as well as other symptoms, may also occur [73,74,75]. Olanzapine is classified as a second-choice drug due to its significant impact on metabolism. Nevertheless, due to its considerable efficacy, it continues to be one of the most commonly used second-generation antipsychotics, along with quetiapine, risperidone, and aripiprazole [41,68,76]. 

#### 2.4.2. Preclinical OLZ Studies

There are numerous findings to compare oxidative stress with OLZ toxicity, including preclinical and clinical studies [77,78,79,80,81]. In numerous projects, olanzapine is compared with other antipsychotics, both first- and second-generation. In the animal study by Reinke et al., it was combined with haloperidol and clozapine. The obtained results indicate a significant advantage of atypical drugs over classical ones. Unlike haloperidol, they did not cause oxidative brain damage in the studied rats. A 28-day treatment with olanzapine administered intravenously at doses of 2.5, 5, and 10 mg/kg resulted in lower TBARS and carbonyl levels. Two-month treatment of rats with schizophrenia (doses of 10–20 mg/day) also resulted in improvements in oxidative stress parameters. Lipid peroxidation was assessed by analyzing MDA concentrations. Compared to pre-treatment values, biomarker levels decreased significantly [79]. 

Research from recent years also indicates an important role of the prefrontal cortex in the treatment of schizophrenia. Olanzapine-induced inflammation of this area and activation of the immune response are most likely linked to the pathogenesis of schizophrenia. Administration of the drug to rats for 1, 8, and 36 days (dose 3 mg/kg/day) resulted in activation of IKKβ/NF-κB inflammatory signaling pathways, increased expression of the pro-inflammatory cytokines TNF-α, IL-6, and IL-1β, and immune system proteins including iNOS, TLR4, and CD14 in the prefrontal cortex [81].

OLZ also shows effects on hypothalamic cells, with dose- and time-dependent effects suggested. During in vitro studies on cell lines, a significant decrease in cell survival was observed at concentrations ≥100 μM after 24 h of incubation. Quantification of live hypothalamic cells confirmed the neurotoxicity of OLZ after both 24 and 48 h of incubation, which was for concentrations of 100 μM. No significant changes in cell survival were observed at a concentration of 25 μM. Based on the results, it can be concluded that olanzapine leads to dose-dependent neurotoxicity, where low concentrations of olanzapine (<100 µM) show beneficial antioxidant properties, but high doses (≥100 µM) result in cell damage [78]. 

#### 2.4.3. Clinial OLZ Studies

Schizophrenic patients have been shown to have elevated levels of free radicals, with reduced levels of antioxidants. However, a significant number of studies show improvements in oxidative stress parameters in the blood of patients treated with OLZ. In the group of patients receiving the drug for 3 months at a dose of 5–20 mg/day, there was a decrease in the elevated MDA levels. The improvement was also noticeable for GSH, the concentration of which decreased significantly. In addition, the initially reduced levels of α-tocopheryl and ascorbic acid increased significantly. In this study, similar effects were obtained for risperidone, but better results in improving levels of oxidative stress biomarkers were obtained in the group of patients treated with olanzapine [77]. 

Total antioxidant status (TAS) was measured in schizophrenic patients. As a result of treatment, the blood levels of patients increased significantly. On the basis of the obtained data, researchers concluded that olanzapine showed a partial ability to regulate elevated levels of oxidative stress in patients with schizophrenia [80]. 

### 2.5. Quetiapine (QT)

#### 2.5.1. Pharmacological QT Data

Quetiapine (QT), like clozapine and olanzapine, belongs to the dibenzoazepine derivatives. In 1997, it was approved by the FDA for the treatment of schizophrenia. In 2004, the registration indication was expanded to include the treatment of bipolar disorder and, a few years later, also the treatment of depression, with quetiapine being recognized as an adjunctive therapy drug [73,82,83]. It is characterized by strong antagonism to the 5-HT_2A_ receptor. It binds to a small extent to the D_2_ receptor. Its sedative effect is most likely related to its very strong affinity for histamine H_1_ receptors. In turn, the occurrence of orthostatic hypotonia is explained by its affinity for α_1_-adrenergic receptors. Although the mechanisms of action of quetiapine are not fully understood, it is indicated that its metabolite—norquetiapine—influences antidepressant efficacy. This compound not only acts agonistically at 5-HT_1A_ receptors but also inhibits norepinephrine transport factor (NET). Side effects include drowsiness, dry mouth, dyslipidemia, and weight gain. However, compared to olanzapine and clozapine, the intensity of metabolic disturbances occurring during quetiapine use is lower [41,68,73,83,84,85]. 

#### 2.5.2. Preclinical QT Studies

Studies have shown that quetiapine presents different directions of action from the damaging effects of oxidative stress [86,87,88,89,90,91]. Han et al. induced free radical production by a seven-day intra-peritoneal administration of 20% ethanol (2 g/kg/day) to rats. An increase in total antioxidant capacity (TAC), SOD, and CAT activity was observed in brain areas of animals receiving the drug at a dose of 10 mg/kg/day. This effect was not observed in the liver, where there was an increase in MDA and ROS concentrations despite drug administration [86]. Similar results were also obtained with fourteen days of quetiapine therapy (20 mg/kg), which produced a significant increase in SOD activity and a decrease in TBARS levels in the rat brain areas studied (hippocampus, amygdala, and prefrontal cortex). This indicates a reduction in lipid peroxidation and protein damage, thereby reducing oxidative stress [87]. Due to the significant effectiveness of quetiapine in the treatment of cognitive deficits, studies were also conducted to assess its impact on behavioral changes resulting from anticancer treatment with doxorubicin. The results of the 30-day study revealed that oxidative stress and inflammation parameters were improved in the brain tissue of the rats. Quetiapine administration reduced high concentrations of MDA and GSH in the brains of rats [91].

#### 2.5.3. Clinical QT Studies

In contrast to preclinical data, determination of myeloperoxidase (MPO) and C-reactive protein (CRP) activity in patients after QT overdose revealed oxidative dysfunction and the development of inflammation. Levels of both biomarkers increased significantly, which was further related to serum drug concentration and the patient’s clinical condition. Quetiapine concentrations above 1 μg/mL in blood correlated with both increased MPO and CRP levels [88]. However, in other studies conducted with quetiapine-treated patients, the antioxidant effect of the drug has been observed. After four weeks of therapy, a decrease in baseline-elevated plasma TBARS and urinary F2-isoprostane concentrations was noted. Initial levels of the lipid peroxidation marker 4-hydroxynonenal (4-HNE) in the schizophrenia group were not significantly different from the control group, and these levels did not change after quetiapine treatment. This study also allowed a comparison of the antioxidant properties of selected atypical antipsychotics. Quetiapine, together with clozapine and olanzapine, showed the most beneficial effect on oxidative stress parameters [89]. 

This effect was also confirmed during in vitro studies using blood drawn from healthy volunteers. Significant reductions in lipid peroxidation in the form of reduced TBARS levels occurred after both 1 and 24 h of plasma incubation with the drug [90]. 

### 2.6. Aripiprazole (ARP)

#### 2.6.1. Pharmacological ARP Data

In 2002, the FDA approved the new drug aripiprazole (ARP) for the treatment of schizophrenia. As a quinoline derivative, it exhibits a unique mechanism of action, making it classed as a third-generation antipsychotic. Like quetiapine, aripiprazole has received approval for the treatment of bipolar affective disorder and depression as adjunctive therapy. This occurred in 2004 and 2007 [73]. It is called a dopamine system stabilizer. When the activity of the dopaminergic system is reduced, it acts as an agonist for D_2_ receptors by stimulating them. In contrast, when dopamine concentrations are too high, aripiprazole acts as an antagonist. This allows for antipsychotic effects while reducing the risk of EPS. This drug also has an affinity for 5-HT_1A_, 5-HT_2A_, 5-HT_7_, and D_3_ receptors and moderate activity at D_4_, H_1_, and α_1_-adrenergic receptors. A particular advantage of aripiprazole is its beneficial effect on prolactin. This is because it does not increase serum concentrations of the hormone. In addition, it can lead to the normalization of prolactin levels in patients whose levels were initially elevated. The most common side effects include drowsiness, akathisia, anxiety, nausea, vomiting, and headache [56,68,73,92,93,94,95,96]. 

#### 2.6.2. Preclinical ARP Studies

Preclinical, in vitro, and clinical studies have shown that ARP can generate or reduce oxidative stress [97,98,99,100,101]. It was observed that in both isolated mouse liver mitochondria and in isolated blood cells from healthy patients, mitochondrial respiration was disrupted by aripiprazole administration. This drug led to the induction of reactive oxygen species production and lipid peroxidation, as indicated by increased levels of modified HNE proteins [97]. 

In contrast, some studies confirm the induction of an antioxidant response by ARP. In view of its potential antidepressant effect, an analysis was made of the drug’s effect on oxidative stress parameters that also play an important role in the pathogenesis of depression. Lipid peroxidation levels in the cerebral cortex of rats receiving aripiprazole (2.5 mg/kg) for four weeks did not differ from those of the control group. However, glutathione peroxidase activity, GSH, and vitamin C concentrations increased significantly [99]. 

Recent studies also indicate a potential antioxidant effect on neurons. Administration of aripiprazole to rats for 30 days at doses of 1 and 2 mg/kg resulted in reduced levels of the oxidative stress marker MDA and the inflammatory marker COX-2 for both drug concentrations. In addition, the antioxidant activities of GSH and CAT increased. According to the researchers, the results obtained demonstrate the neuroprotective mechanism of action of aripiprazole [100]. 

Equally beneficial effects of the drug on antioxidant protection and cell viability were observed in relation to rat hepatocytes. Under oxidative conditions, after eight weeks of treatment, there was an increase in the activity of the antioxidant enzymes, SOD and CAT, which applied to both 2.23 and 6 µM ARP concentrations. At the same time, the dose-dependent effect of the drug was shown, and the largest increase in enzymes occurred after the drug was administered at a higher concentration [101]. 

#### 2.6.3. Clinical ARP Studies

In contrast, the results of a clinical study conducted to assess the effect of therapeutic doses of antipsychotics on oxidative stress parameters suggest that aripiprazole has a dose-dependent effect. Plasma samples from healthy volunteers were incubated with the drug for 24 h. For both drug concentrations of 163 and 242 ng/mL, no significant effect was observed on plasma TBARS levels. However, at lower concentrations, there was a slight induction of lipid peroxidation, indicating possible prooxidant properties [98]. 

## 3. Mood Stabilizers

### 3.1. Lithium (Li)

#### 3.1.1. Pharmacological Li Data

The origins of the use of lithium (Li) date back to the 19th century, but the first publications demonstrating its effectiveness were published in the mid-20th century [102,103]. It is currently used mainly in the treatment of schizoaffective disorders, including bipolar mood disorders, as a mood stabilizer [104]. It is also suggested for use in the treatment of schizophrenia as a next-line treatment and in severe forms of depression due to its anti-suicidal properties [105,106]. Its use in neurodegenerative diseases has also been postulated [107,108,109]. Despite its long history of use, the mechanism of action of lithium is not yet fully understood. However, two main mechanisms of action have been postulated: by blocking inositol phosphatases involved in intracellular signal transduction and by inhibiting glycogen synthase kinase 3 [110,111]. In addition, it increases GABA-based inhibitory neurotransmission, which is of particular importance in the treatment of mania [112]. Side effects associated, particularly with long-term use, include kidney damage and endocrine disruption in the form of deterioration of thyroid functions [113,114].

#### 3.1.2. Preclinical Li Studies

Preclinical, clinical, and in vitro studies have shown that lithium enhances or inhibits oxidative stress [115,116,117,118,119,120,121,122,123]. In preclinical studies conducted on rats induced into a state of mania by methamphetamine, it was shown that the oxidative effect of lithium is influenced by the dose of the substance inducing the manic state in the model and the site of action in the nervous tissue [118]. Results confirming the antioxidant properties of lithium were also shown in a study by Jornad et al., which was also conducted on rats introduced into a model of mania. SOD activity was statistically significantly reduced with lithium treatment compared to the group introduced into mania without lithium treatment in both the prefrontal and hippocampus. In the case of CAT, increased activity was also observed in both brain regions. In addition, a significant reduction in TBARS was also observed after lithium therapy in neural tissue. In contrast, in non-manic animals, the use of lithium did not affect oxidative stress parameters in any case [119]. Another study in rats showed a significant increase in SOD levels only in the group treated with higher concentrations of lithium, suggesting a dose-dependent increase in oxidative damage [121].

Additionally, in an in vitro model study, lithium was shown to inhibit oxidative stress-induced senescence in nerve cells [120]. In a study conducted on rat liver cells in vitro, lithium cytotoxicity associated with its participation in the formation of oxygen-free radicals was demonstrated. Moreover, cytotoxicity increased with increasing ion concentrations. In addition, the role of the CYP2E1 isoenzyme in oxidative stress associated with the intake of lithium was postulated [123]. 

#### 3.1.3. Clinical Li Studies

A clinical study conducted on a group of bipolar disorder patients treated with lithium showed a statistically significant reduction in TBARS level, an exponent of increased lipid peroxidation, after starting therapy. Similar observations have been shown for superoxide dismutase (SOD). CAT and GPx levels did not change statistically significantly after lithium treatment but were significantly higher compared to the control group, which is explained by the pathophysiology of the disease. In addition, changes in oxidative stress parameters have been linked to the response to lithium therapy [115,116]. A study by Banerjee et al., also conducted on a group of bipolar disorder patients, showed significantly reduced lipid peroxidation exponents in lithium-treated patients compared to lithium-naive patients. Moreover, an increase in Na^+^-K^+^-ATPase activity associated with lithium treatment and affecting the reduction of oxidative stress was demonstrated [117]. In contrast, a study in healthy volunteers showed a statistically significant reduction in SOD after Li therapy, with no significant effect on CAT or TBARS levels [122]. 

### 3.2. Valproic Acid (VPA)

#### 3.2.1. Pharmacological VPA Data

The action of valproic acid (VPA) is based on blocking voltage-gated ion channels and modulating GABA-nergic and monoamine-based transmission within neural tissue. Through the induction of glutamic acid decarboxylase, VPA increases levels of GABA, which is an inhibitory neurotransmitter. In addition, VPA inhibits the enzymes that break down GABA, which further contributes to its levels. This reduces impulse conduction, which translates into a reduction in the occurrence of convulsions. The effect on NMDA receptor-based conduction is also not insignificant. This is carried out by weakening the action of glutamate, which is a stimulating neurotransmitter. In addition, VPA reduces glucose utilization within the brain, which results in reduced ATP production and indirectly translates into a slower metabolism in the neural tissue. Common side effects associated with the use of VPA include weakness, lethargy, nausea, weight gain, and hair condition deterioration. In addition, hematological disorders and hepatotoxicity associated with VPA intake are observed. In addition, valproic acid has shown significant teratogenic properties [124,125,126,127,128,129,130,131,132,133,134,135,136]. 

#### 3.2.2. Preclinical VPA Studies

The use of valproic acid affects the redox balance in the body [137,138,139,140,141,142,143,144,145]. Studies in rats have shown VPA-associated significant reductions in GTx, SOD, and CAT activity in neural tissue. In addition, oxidative stress exponents such as TBARS and carbonyls have been shown to be elevated [137]. However, this effect can be inhibited by selenium and L-carnitine supplementation, which has also been confirmed in studies conducted on rats [138,139]. In turn, work carried out on the same animal model showed neuroprotective properties of VPA against nerve cells after the occurrence of mechanical trauma in the mechanism of inhibiting oxidative stress [140]. VPA also has a similar effect in the case of nerve tissue damage as a result of stroke, which has been demonstrated in studies on the same animal model [141]. 

Studies conducted on human neuroblastoma cells in an in vitro model showed a reduction in hydrogen peroxide levels in cells subjected to stimulated excitotoxicity in the case of incubation with VPA. In addition, an increase in SOD and a slight increase in CAT were also observed in samples incubated with VPA. However, when excitotoxicity was induced, the relationships were not statistically significant [142]. Also, a study conducted on a mouse embryo model showed an increase in reactive oxygen species associated with VPA administration and a negative impact on the embryo. In addition, administration of exogenous CAT showed potential embryonic protective properties, which could not be demonstrated for SOD [144]. Elevated markers of oxidative stress associated with the use of VPA have also been demonstrated in work on glioma cell lines. VPA-related reactive oxygen species caused damage within mitochondria, affected cell–cell communication pathways, and regulated VPA-induced apoptosis [145]. 

#### 3.2.3. Clinical VPA Studies

A clinical study conducted on a group of VPA-treated patients showed a significant reduction in antioxidant potential (FRAP) between the VPA-treated and the control group. Importantly, no such relationship was shown for reduced to oxidized glutathione (GSH/GSSG ratio) [143]. 

### 3.3. Carbamazepine (CBZ)

#### 3.3.1. Pharmacological CBZ Data

The origins of the use of carbamazepine (CBZ) date back to the mid-20th century. Its mechanism is based on the modulation and blocking of potential-gated sodium channels. Its effects on serotonin- and catecholamine-based conductance have also been postulated. Among its other uses are the treatment of seizures, neuropathic pain, and manic episodes in the course of bipolar disorder. Side effects associated with its use include feelings of fatigue, headaches, blurred vision, and ataxia. Cases of Stevens–Johnson syndrome, severe hematological disorders, and toxic epidermal necrolysis associated with carbamazepine use have also been described [146,147,148,149,150,151,152]. 

#### 3.3.2. Preclinical CBZ Studies

The studies of the effect of CBZ on the condition of oxidative stress are varied [153,154,155,156,157,158,159,160]. In vitro studies on erythrocytes showed no significant effect of the drug on oxidative stress in the cell [155]. In contrast, a study conducted on a plant cell model showed an increase in lipid peroxidation and levels of hydroperoxide and carbonyl proteins. In addition, a statistically non-significant increase in CAT levels was observed [160].

Studies of CBZ nephrotoxicity in rats suggest an association of renal cell damage with increased rates of oxidative stress [156]. In another study conducted on rats, no significant changes in malondialdehyde, glutathione, and SOD levels were observed in neural tissue during CBZ treatment. A significant decrease in GSH and SOD levels and an increase in MDA levels were only visible in the group with induced epilepsy [157]. A study in another animal model showed an increase in lipid peroxidation and hydroperoxide levels after exposure to CBZ. At the same time, a reduction in SOD, CAT, and GPx activity was confirmed [158]. Similar conclusions were reached in a similar study on the same animal model. A reduction in SOD, CAT, and GPx activity was shown with a concomitant increase in TBARS and carbonyl protein [159]. 

#### 3.3.3. Clinical CBZ Studies

A study conducted on a pediatric population showed a significant reduction in antioxidant capacity and an increase in the oxidative stress index among epilepsy patients treated with carbamazepine compared to a healthy control group. In addition, an association was found between the use of CBZ and increased total oxidative status [160]. Similarly, a study by Varoglu et al. found a significant reduction in some antioxidative markers, with an increase in selected oxidative markers during carbamazepine monotherapy [154]. 

### 3.4. Oxcarbazepine (OXC)

#### 3.4.1. Pharmacological OXC Data

The origins of the use of oxcarbazepine (OXC) in therapy date back to the 1990s. It is a structural derivative of carbamazepine, which is a pro-drug that is activated in the body as an active metabolite. It is postulated to act mainly by blocking and influencing voltage-gated sodium channels. The primary uses of the drug include the treatment of epilepsy and affective disorders. Compared to carbamazepine, OXC is characterized by greater safety and fewer side effects and interactions. Common side effects associated with OXC use include dizziness, headaches, and nausea [161,162,163,164,165,166]. 

#### 3.4.2. Preclinical OXC Studies

The studies of the effect of OXC on the status of oxidative stress are not very numerous and clear [167,168,169,170,171]. Studies in mice have shown no significant effect of OXC use on levels of oxidative stress exponents such as MDA, GSH, CAT, and SOD determined in brain tissue. Elevated MDA levels and reduced glutathione levels were only observed with OXC use in individuals with pharmacologically induced epilepsy [168]. In turn, studies conducted in rats showed a significant protective effect of the use of OXC on nerve cells after hypoxia induced by cardiac arrest. It is postulated that these protective properties result from the attenuation of oxidative stress by increasing the presence of SOD [169]. The neuroprotective properties of OXC through its effect on redox balance are also confirmed by a rodent study by Park et al., which showed a reduction in the production of superoxide anions. 

In addition, elevated levels of SOD, CAT, and GPx were observed in nerve cells [170]. An increase in the formation of ROS as a result of epileptic seizures has been demonstrated [171].

#### 3.4.3. Clinical OXC Studies

A meta-analysis by Rezaei et al. found no significant statistical difference between OXC-treated and healthy individuals in blood-assessed homocysteine levels, which are directly related to redox balance regulation [167]. Similar observations have also been made regarding folic acid and cobalamin’s indirect involvement in the regulation of oxidative stress [168].

## 4. Antidepressants

### 4.1. Citalopram (CIT) 

#### 4.1.1. Pharmacological CIT Data

Citalopram (CIT) is a selective serotonin reuptake inhibitor (SSRI), authorized in 1998. In addition, it has a low affinity for dopaminergic D_2_ receptors, 5-HT_1A_ and 5-HT_2A_ receptors, α-adrenergic and β-adrenergic, muscarinic, and histamine H_1_ receptors [172]. It is used in the treatment of depressive disorders, anxiety disorders with or without anxiety attacks with agoraphobia, as well as obsessive-compulsive disorders. Common side effects include sleep disturbances, nausea, decreased sex drive, excessive sweating, and dry mouth [173]. 

#### 4.1.2. Preclinical CIT Studies

The research results provide a lot of data on the impact of CIT on oxidative stress [174,175,176,177,178,179,180,181]. Hepatotoxicity of CIT, resulting from free radical effects, has been observed in rat studies. Treatment took place in vitro and in vivo. In the in vitro group, hepatocyte death was noted at 500 µM, as well as increased ROS formation, breakdown of mitochondrial potential, lysosomal membrane leakage, GSH depletion, and lipid peroxidation. The in vivo tests at a dose of citalopram (20 mg/kg), confirmed the above damage [174]. 

In addition, other studies conducted on male rats have shown that CIT causes testicular damage to the mechanisms of oxidative stress as well as hormonal changes. The drug was used at doses of 5, 10, and 20 mg/kg. Sperm showed abnormal morphology and DNA damage, while their concentration decreased. In addition, histopathological changes in the testicles were observed. Luteinizing hormone concentrations increased with each dose of citalopram, while testosterone concentrations increased with doses of 5 and 10 mg/kg. Reduced glutathione levels signaled increased oxidative stress in the male rats given citalopram at doses of 10 and 20 mg/kg [175]. Studies conducted on Daphnia magna have shown that citalopram causes oxidative damage. The microorganisms were exposed to citalopram in the water at a concentration of 1.03 mg/L. In Daphnia, an increase in the concentration of the antioxidant molecule glutathione S-transferase (GST) was observed, as well as an increase in the activity of SOD, CAT, and GPx. In addition, MDA, protein carbonyl, and 8-OHdG concentrations increased. CIT has been shown to cause oxidative damage [176]. 

Antitumor effects of citalopram on hepatocellular carcinoma cells have also been observed. The cytotoxic effect of the drug on HepG2 cells was manifested by decreased cell viability and increased ROS formation. An increase in mitochondrial Bax and a decrease in Bcl2, as well as cytochrome C release, were observed. It has been suggested that citalopram may exhibit apoptotic effects against a hepatocellular carcinoma cell line via induction of cell death via a cytochrome C release mechanism and ROS-dependent activation of NFκB [177]. 

In addition, studies on the effect of citalopram on the processing of amyloid-β precursor protein (AβPP) have suggested that the drug may increase AβPP processing and also reduce oxidative stress. The study was conducted on induced pluripotent stem cells (iPSCs). Citalopram was administered for 45 days at concentrations of 0.8, 5, and 10 µM. In cells with the *PSEN1* gene mutation, O_2_ concentrations were significantly reduced after each dose of citalopram. In contrast, no change in the formation of terminal oxidative stress markers was observed after treatment [180].

In addition, in studies conducted in rats, it was observed that co-administration of citalopram combined with rosuvastatin intensifies oxidative stress. The concomitant medications were used for 14 days. The results showed an increase in peroxidase and glutathione reductase activity, with no effect on antioxidant concentrations [178]. On the other hand, in studies conducted in Wistar rats, a neuroprotective effect of citalopram was observed, induced by the alleviation of oxidative stress, inflammation, apoptosis, and an altered metabolic profile. Treatment with CIT at 4 mg/kg significantly altered 17 metabolites with attenuation of malondialdehyde, reduced glutathione, matrix metalloproteinases, and apoptosis markers [179]. 

#### 4.1.3. Clinical CIT Studies

A limited number of studies examine oxidative stress measured in patient groups. There was a significant increase in serum SOD, serum MDA, and decrease in plasma ascorbic acid levels in patients treated with CIT for major depression as compared to control subjects [181].

### 4.2. Sertraline (SER) 

#### 4.2.1. Pharmacological SER Data

Sertraline (SER) was introduced in 1991. It is a naphthaleneamine derivative. The mechanism of action of the drug is to inhibit the presynaptic reuptake of serotonin from the synaptic cleft (SSRI). Norepinephrine and dopamine transport are also blocked to a small extent. Sertraline slightly binds to adrenergic, histamine, muscarinic, dopaminergic, serotonergic, benzodiazepine, and GABA receptors. It is used to treat depressive disorders, obsessive-compulsive disorder, post-traumatic stress disorder, and seizure disorders. Common side effects include insomnia, drowsiness, anorexia, tremors, dizziness and headaches, diarrhea, nausea, or sexual dysfunction. In addition, movement disorders, paresthesias, visual disturbances, or increased sweating have been noted. In addition, sertraline-induced hepatotoxicity has also been described [182,183,184]. 

#### 4.2.2. Preclinical SER Studies

The effects of SER on lipoperoxidation and antioxidant enzyme activity in plasma and brain tissues were studied in Wistar albino rats. Animals were administered the drug at doses of 10, 40, and 80 mg/kg. Lipid peroxidation (MDA) levels in plasma and brain tissue increased in all rats treated with sertraline. SOD activity in brain tissue decreased, while plasma concentrations increased compared to the control group. In contrast, plasma and brain tissue CAT concentrations and plasma paraoxonase (PON) concentrations decreased compared to the control group. Administration of higher doses of sertraline has been found to increase oxidative stress [185]. Studies in Drosophila melanogaster have shown sertraline-induced DNA damage and cell toxicity. Antidepressant-treated larvae manifested delayed development and reduced survival. It was observed that, through the action of sertraline, mitotically active tissues had DNA breaks and underwent apoptosis with increased frequency. Interestingly, the toxicity of the drug was partially mitigated by the administration of ascorbic acid. It has been suggested that sertraline induces oxidative DNA damage [186]. Sertraline-induced hepatotoxicity is probably due to mitochondrial dysfunction. The study was conducted on male Sprague Dawley rats. It has been observed that at doses of 75 and 100 µM, sertraline inhibits mitochondrial function by uncoupling oxidative phosphorylation and inhibiting the activity of complexes I and V. This process is further mediated by the loss of ATP and the induction of microsomal TG transport protein (MPT) [184]. 

The effects of sertraline (also in combination with haloperidol) on oxidative stress in mice were also compared. SER was administered at doses of 10–20 mg/kg. In antidepressant monotherapy, brain concentrations of GSH, MDA, TAC, and nitrite remained unchanged, while catalase and PON1 activity decreased. It has been found that sertraline, through its effects, can expose the brain to further oxidative damage [187]. Importantly, some studies suggest using the oxidative properties of sertraline to treat certain diseases. Sertraline can be used in the treatment of prostate cancer. Antidepressant-induced apoptosis and autophagy by activating free radical production, hydrogen peroxide (H_2_O_2_) formation, lipid peroxidation, and decreasing GSH concentration. Furthermore, sertraline significantly reduced the expression of aldehyde dehydrogenase 1 (ALDH1) and the stem cell marker CD44 [188]. 

Other studies have investigated the oxidative properties of sertraline against the Leishmania infantum parasite. The drug-induced uncoupling of respiration, a significant decrease in intracellular ATP concentration, and oxidative stress in the promastigote *L. infantum*. In addition, prolonged metabolic dysfunction was demonstrated. Sertraline killed *Leishmania* through a multidirectional mechanism of action, eliminating the parasite’s primary metabolic pathways. The use of sertraline to treat visceral leishmaniasis in an off-label regimen has been suggested [189]. 

#### 4.2.3. Clinical SER Studies

In contrast, a pilot study comparing the effects of sertraline on markers of oxidative stress (MDA) revealed a different effect on oxidative properties. People with depression, also diagnosed with heart failure, were divided into two groups: those receiving sertraline and those not receiving any antidepressant. After three months of treatment, patients treated with sertraline showed a significant reduction in MDA. It has been suggested that SER treatment reduces plasma markers of oxidative stress in depressed patients with additionally diagnosed heart failure [190].

### 4.3. Venlafaxine (VEN) 

#### 4.3.1. Pharmacological VEN Data

Venlafaxine (VEN) was launched in 1994 [191]. This substance, together with its metabolite—O-desmethylvenlafaxine—acts as a serotonin and norepinephrine reuptake inhibitor (SNRI). In addition, venlafaxine shows weak properties as a dopamine reuptake inhibitor. Its serotonergic effect contributes to sedative, drowsiness-increasing, and neutralizing effects, while its noradrenergic action can result in the opposite effects, i.e., agitation, deterioration of sleep quality, and sometimes even induce feelings of anxiety. The drug is approved for a variety of uses, such as the treatment of episodes of major depression, prevention of recurrent episodes of major depression, treatment of generalized anxiety disorder, treatment of social phobia, and treatment of paroxysmal anxiety [192]. Most patients tolerate VEN well, although symptoms such as dizziness and headache, nausea, dry mouth, and night sweats are most commonly seen in clinical practice. An increase in blood pressure is also observed, especially at the beginning of therapy or with high daily doses (300 mg/day) [193]. Other potential side effects mentioned in the literature include tinnitus, palpitations, sleep disturbances, vomiting, diarrhea, constipation, decreased appetite, and nervousness [194].

#### 4.3.2. Preclinical VEN Studies 

A study in male mice assessed venlafaxine’s ability to damage DNA in the brain and liver and its oxidative effects on DNA, lipids, and proteins. An intermediate dose of the drug (50 mg/kg) caused significant DNA damage at 2 and 6 h after exposure, while a high dose (250 mg/kg) caused even more pronounced oxidative damage over the same time interval. Damage was observed in both the brain and liver, with the liver being, however, more severe. Lipoperoxidation effects in the brain and liver were evident after 6 and 12 h at doses of 50 and 250 mg/kg. In contrast, liver nitrite content induced 2 h after exposure to venlafaxine (250 mg/kg) was elevated [195]. 

Similar observations were made on the effect of VEN on hepatocyte cytotoxicity—in this study, the drug was administered to male rats, and then liver cells were isolated from them. Venlafaxine has been observed to cause hepatotoxicity through the induction of oxidative stress and subsequent toxic effects, including GSH depletion, lipid peroxidation, breakdown of mitochondrial potential, and lysosomal membrane leakage in hepatocytes. In addition, functional mitochondrial damage was observed through inhibition of mitochondrial respiratory complexes II and IV. Lysosomes and mitochondria have been shown to be the earliest target structures for venlafaxine-induced hepatotoxicity [196]. 

In addition, chronic VEN administration affects the methylation of promoters of genes involved in oxidative and nitrosative stress. This includes mRNA expression of SOD1, SOD2, and NOS1 in peripheral blood mononuclear cells (PBMC) and in the brain, as well as gene expression of CAT, Gpx1, and Gpx4 only in the brain [197]. In a study in mice, co-administration of VEN (40 mg/kg) with morphine showed inhibition of naloxone-induced withdrawal symptoms. In addition, the drug reduces the expression of TNF-α, IL-1β, IL-6, NO, and the concentration of MDA in brain tissue. Repeated administration of venlafaxine inhibits the decline in brain-derived neurotrophic factor (BDNF), thiol, and GPx [198]. Studies evaluating the effects of venlafaxine on glucose homeostasis and oxidative stress showed that the drug’s effects were associated with a TBARS decrease and an increase in GSH levels in the brain. Mice with diabetes were studied; VEN was administered at doses of 8 and 16 mg/kg/day for 21 days. Glucose levels did not change significantly after administration [199]. No clinical VEN study results were found in the literature. 

## 5. Discussion

Understanding the mechanisms and role of oxidative stress in the possible etiopathogenesis of psychiatric diseases makes it possible to develop new therapeutic approaches based on its modulation. Such research has been conducted in the case of bipolar disorder, where oxidative stress has been shown to affect molecule damage at the neurobiological level. This observation makes it possible to try to use drugs that affect the balance between antioxidative and oxidative processes to induce remission of the disease [1].

A compilation of the many publications on in vitro and animal experiments and the few findings from studies conducted in patient groups indicates considerable interest in the topic of the involvement of oxidative stress in the mechanism of many neurodegenerative diseases treated with older and newer generation drugs. 

The findings indicate that significant differences can be observed in clinical practice regarding the prescribing of the classic drug haloperidol. Although second-generation drugs are preferred in most cases, haloperidol still stands out among the classic antipsychotics, as its prescription frequency has not changed significantly over the past decade [41,42,43]. Despite this, the contribution of oxidative stress during HLP use is still not fully understood. A substantial body of research allows the authors to conclude that haloperidol increases oxidative stress in brain tissue, showing a tendency to increase neuronal damage [47]. Hence, one of the main directions of recent research is the study of the interaction of antipsychotic drugs, including haloperidol, with other substances. This is because they can exhibit both adverse and beneficial joint effects on oxidative stress parameters [44,47,51]. 

Continued research among neuropsychiatric drugs has confirmed the significant efficacy of clozapine in the treatment of symptoms of psychosis, particularly drug-resistant schizophrenia [52,53]. Studies on the involvement of clozapine in oxidative stress further revealed the cardiotoxic effects of the drug, confirmed by elevated activities of biomarkers of cardiac damage, i.e., LDH and cardiac creatine kinase isoenzyme, CK-MB, with a concomitant decrease in GSH levels, such an important antioxidant in the body [58]. However, the results of a study comparing oxidative stress parameters in patients treated with perphenazine, risperidone, and clozapine led the authors to a completely different conclusion. Indeed, it was shown that in the group of patients treated with clozapine for at least three months, serum GSH and SOD levels increased, and MDA levels were significantly lower compared to patients treated with perphenazine and risperidone. It has, therefore, been suggested that clozapine, contributes to a significant alleviation of the negative symptoms of schizophrenia by exhibiting antioxidant effects [61]. 

The results suggest that atypical antipsychotics (clozapine, risperidone) have a more favorable effect on levels of oxidative stress biomarkers compared to classical drugs (perphenazine), which may be important in the context of both schizophrenia treatment and symptom relief [61]. The success of clozapine led to a search for its chemical analog, which would not be associated with such a high risk of developing agranulocytosis. This research allowed the synthesis of olanzapine, a new tricyclic dibenzoazepine derivative belonging to the second generation of antipsychotics. The drug quickly gained popularity after its approval and, along with risperidone, became one of the most widely used antipsychotics. Its prescribing frequency declined slightly in the 2000s, most likely driven by patient concerns about side effects [41,68,76]. 

Numerous studies have been conducted to clarify the relationship between OLZ treatment and changes in antioxidant and oxidant levels in the body [77]. Given the lack of oxidative damage induced by olanzapine, researchers point to a possible neuroprotective mechanism of action of the drug [79]. The results suggest that the use of olanzapine in the short term, unfortunately, leads to the development of inflammation in both the central and peripheral nervous system, which induces the occurrence of altered immunological responses in patients with schizophrenia. These changes may be related to the therapeutic effects of olanzapine, but their association with adverse side effects, e.g., weight gain, is also not excluded [81]. 

QT shows a complex effect on oxidative stress. As with the other second-generation drugs, research results indicate their antioxidant properties [86,87]. The authors pointed to the potential use of quetiapine to improve cognition and reduce oxidative damage to the brain, drawing additional attention to the drug’s neuroprotective properties. Indeed, a reduction in Bax and caspase-3 protein levels provides protection against neuronal apoptosis [91,96]. The favorable side-effect profile makes aripiprazole one of the drugs preferred by patients. Along with risperidone, quetiapine, and olanzapine, it is the most commonly prescribed atypical antipsychotic drug [56,68,96]. Given the unique mechanism of action of aripiprazole and its favorable side-effect profile, it is expected to have an equally favorable effect on oxidative stress parameters. However, the studies conducted do not provide a clear answer regarding the potential pro- or antioxidant effect of the drug [97]. Recent studies also indicate a potential antioxidant effect on neurons. According to the researchers, the results obtained demonstrate the neuroprotective mechanism of action of aripiprazole [100]. Based on the results, it was inferred that aripiprazole provides potential protection for liver cells under oxidative conditions found in people with schizophrenia [101]. 

The effect of lithium on oxidative status is still not fully understood. A number of factors have now been postulated to influence the effect of lithium on oxidative stress. Among them, factors related to the conduct of therapy should be distinguished, such as the doses used, the time of taking, or interactions with other drugs used. The type of tissue affected by lithium also has an impact. Furthermore, when assessing the effect of lithium on oxidative stress, the pre-existing redox balance disturbances, which form the pathophysiological basis of diseases, must also be taken into account [114,115]. A study in healthy volunteers showed a statistically significant reduction in SOD after lithium therapy, with no significant effect on CAT or TBARS levels [122]. In a study conducted on rat liver cells in vitro, Li cytotoxicity was demonstrated to be associated with participation in the formation of oxygen-free radicals. Moreover, cytotoxicity increased with increasing blood lithium concentrations. In addition, the role of the CYP2E1 isoenzyme in oxidative stress associated with the intake of lithium was postulated [123]. 

Valproic acid and carbamazepine were among the first drugs to be used in the treatment of epilepsy. Toxicity related to the oxidative stress effects of VPA is also seen in rat hepatocyte cells, which is related to the intensive and multi-step hepatic metabolism of the drug [139]. 

Capacity and increased oxidative stress index among epilepsy patients treated with carbamazepine compared to a healthy control group. In addition, a correlation was found between the use of carbamazepine and increased total oxidative status [153]. 

The effect of oxcarbazepine on oxidative stress remains a subject of research [166]. Also, in the case of epilepsy, the role of oxidative stress as one of the components in epileptogenesis has been demonstrated [171]. 

The effect of citalopram on the development of oxidative stress is important. Studies on Daphnia magna have shown that citalopram causes oxidative damage [176]. Antitumor effects of citalopram on hepatocellular carcinoma cells have also been observed. It has been suggested that citalopram may exhibit apoptotic effects against a hepatocellular carcinoma cell line via induction of cell death via a cytochrome C release mechanism and ROS-dependent activation of NFκB [177]. 

The topic of sertraline-induced oxidative stress is frequently addressed in research. Studies in Drosophila melanogaster have shown sertraline-induced DNA damage and cell toxicity. Sertraline-induced hepatotoxicity is probably due to mitochondrial dysfunction [184]. Significantly, some studies suggest using the oxidative properties of sertraline to treat certain diseases. Sertraline can be used in the treatment of prostate cancer [188]. In contrast, a pilot study comparing the effects of sertraline on markers of oxidative stress (MDA) revealed different effects on oxidative properties [190]. 

The relationship between venlafaxine and oxidative stress is a topic addressed in the literature. Studies evaluating the effects of venlafaxine on glucose homeostasis and oxidative stress showed that the drug’s effects were associated with a decrease in TBARS and an increase in brain GSH levels [198].

## 6. Conclusions

In conclusion, oxidative stress is an important risk factor for many psychiatric disorders in the treatment of which first- and second-generation antipsychotics are most commonly used, mood stabilizers, and antidepressants, and this publication reviews the state of the art of the contribution of oxidative stress to the most commonly used treatments of neuropsychiatric diseases, with first- and second-generation antipsychotics as important in treatment success as discoveries in the pathophysiology of neuropsychiatric diseases, including those initiating a new round of innovations in the contribution of oxidative stress to the etiology of neuropsychiatric diseases. For neurological patients, it is important to identify the cause of the oxidation–reduction disturbances, which can be led by pharmacological drug properties, mechanism of action, drug–drug interaction, dose, and nutritional status [2,6,8,77,99,138,139,167,168,181,200]. All changes should be taken into account in oxidative stress generation because it is necessary to understand the characteristics and scope of each neuropsychiatric drug used in therapy.

## Figures and Tables

**Figure 1 ijms-25-07304-f001:**
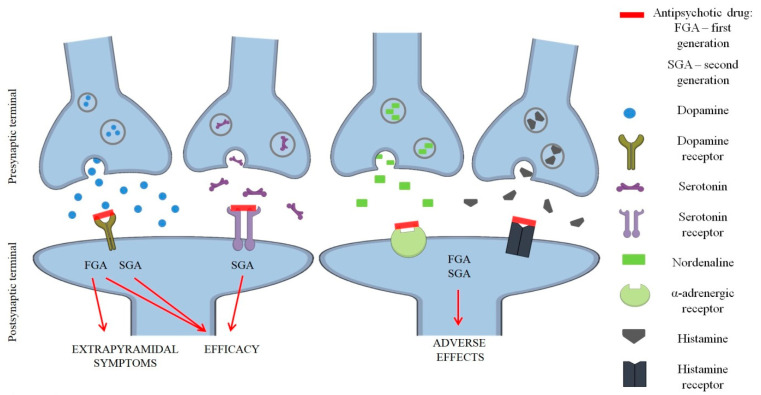
First (FGA) and second-generation (SGA) antipsychotics and extrapyramidal adverse effects—receptor scheme and mechanism [24,25,26].

**Table 1 ijms-25-07304-t001:** Mental and behavioural disorders in selected categories with clinical details [3,4,5,6,7,8,9,10,11,12,13,14,15,16,17,18,19,20,21,22,23].

Mental and Behavioural Disorders	Selected Categories	Clinical Details	Publications
Organic, including symptomatic, mental disorders	Dementia in Alzheimer’s disease	Progressive, neurodegenerative disease Loss of cognitive function	[3,4]
	Vascular dementia	Result of infraction of the brain due to vascular disease	
	Delirium, not induced by alcohol and other psychoactive substances	Non-specific organic cerebral syndrome Including: disturbances of consciousness, attention, perception and emotional disturbances	
	Unspecified organic or symptomatic mental disorder	Brain disorder, injury, or toxicity Mental and behavioural disorders	
Mental and behavioural disorders due to psychoactive substance use	Mental and behavioural disorders due to use of alcohol	Wide variety of disorders resulting from abuse or misuse of alcohol/psychoactive substances	[5,6,7,8]
	Mental and behavioural disorders due to use of sedatives or hypnotics		
	Mental and behavioural disorders due to use of other stimulants, including caffeine		
	Mental and behavioural disorders due to multiple drug use and use of other psychoactive substances		
Schizophrenia, schizotypal, and delusional disorders	Schizophrenia	Multifactorial, combination of genetic and environmental factors Abnormalities in the perception or expression of reality	[9,10,11]
	Schizotypal disorder	Eccentric thoughts, inappropriate affect and behaviour, extreme social anxiety, and limited interpersonal interaction	
Mood (affective) disorders	Manic episode	Extreme changes in behaviour, and mood that drastically affect their functioning	[12,13]
	Bipolar affective disorder	Severe mood swings (mania and depression)	
Neurotic, stress-related and somatoform disorders	Phobic anxiety disorders	Strong, irrational fear, anxiety	[14,15]
	Obsessive-compulsive disorder	Intrusive ideas, thoughts and images	
Behavioural syndromes associated with physiological disturbances and physical factors	Eating disorders	Physiological and psychological disturbances in appetite or food intake	[16,17,18]
	Nonorganic sleep disorders	Disturbance of normal sleep patterns	
	Abuse of non-dependence-producing substances	The use of non-addictive drugs, particularly those purchased without a prescription (antacids, herbal, folk remedies, steroids, hormones and vitamins)	
Disorders of adult personality and behaviour	Specific personality disorders	Disorders characterized by disturbances in the personality	[19,20,21]
	Gender identity disorders	Incongruence between experienced or expressed gender and the one assigned at birth	
	Disorders of sexual preference	Intense sexually arousing fantasies, urges, or behaviours	
Behavioural and emotional disorders with onset usually occurring in childhood and adolescence	Hyperkinetic disorders	Difficulty sustaining cognitive engagement in activities	[22,23]
	Tic disorders	Recurrent tics (involuntary, rapid, nonrhythmic motor movement or vocal)	

**Table 2 ijms-25-07304-t002:** Antipsychotic, mood stabilizers and antidepressants drugs (SSRI, selective serotonin reuptake inhibitor) and (SNRI, serotonin and norepinephrine reuptake inhibitors)—generations and applications [1,3,9,24,25].

Antipsychotic Drugs					Mood Stabilizers				Antidepressants		
Schizophrenia					Bipolar Disorder				Depression		
I	II	III	First-Line Drugs	Second-Line Drugs	I	II	First-Line Drugs	Add-on Therapy	SSRI	SNRI	First-Line Drugs
Haloperidol	Clozapine Quetiapine Risperidone Olanzapine	Aripiprazole	Risperidone Olanzapine Quetiapine Aripiprazole	Clozapine	Lithium Valproate Carbamazepine	Oxcarbazepine	Lithium Valproate Carbamazepine	Oxcarbazepine	Sertraline Citalopram	Venlafaxine	Sertraline Citalopram Venlafaxine

**Table 3 ijms-25-07304-t003:** Biological markers of oxidative stress [31,33,34].

Destructive Changes	Markers of Oxidative Stress
Lipid peroxidation	Malondialdehyde F2-isoprostanes Oxidized low density lipoproteins Oxidized LDL (low-density lipoprotein) antibodies 4-hydroxynonenal Acrolein
Oxidation of nucleic acids	8-hydroxy-2-deoxyguanosine Reactive aldehydes Reduced sugars, e.g., ribose
Carbohydrate oxidation	3-nitrotyrosine
Protein oxidation	Substances that react with thiobarbituric acid Glycation end products Oxidized thiol groups

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
