# Peer review of "The Effect of Neuropsychiatric Drugs on the Oxidation-Reduction Balance in Therapy"

_ijms, 2024, doi:10.3390/ijms25137304_

Round 1

Reviewer 1 Report

Comments and Suggestions for Authors

The review manuscript titled “The Effect of Neuropsychiatric Drugs on the Oxidation-Reduction Balance in Therapy” is devoted to the description of the relationship between psychiatric disorders, available pharmacotherapy and oxidative stress. In the manuscript Authors described  results obtained from clinical and preclinical studies concerning the above mentioned topic. The topic of the manuscript is interesting, but the manuscript requires a few structural improvements.

  1. I believe that the  manuscript should be carefully checked by an English native speaker. Some sentences have strange grammatical constructions and therefore these sentences are difficult to read and understand. Moreover, sometimes  the flow of the text is seriously disturbed. Additionally, I also found a few typos in the text. 

  2. In the manuscript Authors present the effects of selected psychiatric drugs on oxidative stress parameters. The division of this part of the text is only related to the pharmacological classes of drugs. In other words, in the description of particular drugs  after a short introduction Authors mix data from clinical and preclinical studies. Sometimes the reader does not know if Authors describe results from clinical or preclinical studies. It makes the manuscript  chaotic and substantially decreases the level of understanding.  Therefore each chapter (understood as the description of each single drug) should be divided into three different subdivisions. For instance: a) brief pharmacological characteristic b) preclinical studies c) clinical studies. 

  3. I believe that table 1 is completely unnecessary and adds nothing important to the manuscript, therefore it  should be removed.

Comments on the Quality of English Language

I believe that the  manuscript should be carefully checked by an English native speaker. Some sentences have strange grammatical constructions and therefore these sentences are difficult to read and understand. Moreover, sometimes  the flow of the text is seriously disturbed. Additionally, I also found a few typos in the text.

Author Response

Reviewer comment: I believe that the manuscript should be carefully checked by an English native speaker. Some sentences have strange grammatical constructions and therefore these sentences are difficult to read and understand. Moreover, sometimes the flow of the text is seriously disturbed. Additionally, I also found a few typos in the text. 

Response: Thank You for pointing this out. Before being submitted, the publication was translated by a professional company translating medical texts and checked by an English native speaker, but thanks to your attention, the whole thing was sent to them again and checked once more. All grammatical changes and minor language corrections are marked in red.

Reviewer comment: In the manuscript Authors present the effects of selected psychiatric drugs on oxidative stress parameters. The division of this part of the text is only related to the pharmacological classes of drugs. In other words, in the description of particular drugs after a short introduction Authors mix data from clinical and preclinical studies. Sometimes the reader does not know if Authors describe results from clinical or preclinical studies. It makes the manuscript chaotic and substantially decreases the level of understanding. Therefore, each chapter (understood as the description of each single drug) should be divided into three different subdivisions. For instance: a) brief pharmacological characteristic b) preclinical studies c) clinical studies. 

Reviewer comment: I believe that table 1 is completely unnecessary and adds nothing important to the manuscript, therefore it should be removed.

Response: Thank you for this comment, but due to the nature of the content in the table 1 regarding the variety of mental and behavioural disorders, we decided to leave this table, if only because the readers of the work may be specialists in fields other than neurology or psychiatry, so it is good to have a short summary with clinical data and publications where we can find more data.

Reviewer comment: Comments on the Quality of English Language

I believe that the manuscript should be carefully checked by an English native speaker. Some sentences have strange grammatical constructions and therefore these sentences are difficult to read and understand. Moreover, sometimes the flow of the text is seriously disturbed. Additionally, I also found a few typos in the text.

Response: Thank You for pointing this out. All grammatical changes and minor language corrections were corrected.

Reviewer 2 Report

Comments and Suggestions for Authors

Well written paper, I am not a specialist on neuropsychiatry, but on nutrition - and as many patients with mental illnesses have nonoptimal food intake, with vitamin deficiencies (most evident with B and folate deficiencies), but also deficient intake to boost eg.  dopamin and serotonin, I miss a discussion on this. Some paragraphs on this would be nice. See reference: Lee H. The Importance of Nutrition in Neurological Disorders and Nutrition Assessment Methods. Brain Neurorehabil. 2022 Mar;15(1):e1. https://doi.org/10.12786/bn.2022.15.e1

A table 4 with most important drug effects on oxidation-reduction balance would enhance the scientific message. The text is complex with a lot of data and not always easy to read.

The abstract should focus on main results of the review, this also holds for the conclusion

Author Response

Reviewer comment: Well written paper, I am not a specialist on neuropsychiatry, but on nutrition - and as many patients with mental illnesses have nonoptimal food intake, with vitamin deficiencies (most evident with B and folate deficiencies), but also deficient intake to boost eg.  dopamin and serotonin, I miss a discussion on this. Some paragraphs on this would be nice. See reference: Lee H. The Importance of Nutrition in Neurological Disorders and Nutrition Assessment Methods. Brain Neurorehabil. 2022 Mar;15(1):e1. https://doi.org/10.12786/bn.2022.15.e1

Reviewer comment: A table 4 with most important drug effects on oxidation-reduction balance would enhance the scientific message. The text is complex with a lot of data and not always easy to read.

Response: Undoubtedly, this comment is very valuable, although we tried to prepare such a large table collecting all the data, but unfortunately it turned out to be too extensive. To make the text easier to read, we propose a detailed study of individual drug descriptions, highlighting the pharmacological data of the drug and the results of preclinical and clinical studies. I hope this makes reading easier.

Reviewer comment: The abstract should focus on main results of the review, this also holds for the conclusion

Response: The abstract was changed and reviewed.

Reviewer 3 Report

Comments and Suggestions for Authors

This study aims to investigate the relationship between neuropsychiatric drugs and oxidative stress, a potential factor in several mental illnesses. The authors suggest that a better understanding of oxidative stress and its interaction with drugs may lead to more effective therapies.

Some suggestions:

- Paragraphs on oxidative stress (sentences 68-102): Shorten the text by briefly defining oxidative stress and its potential role in various health conditions, including mental illness. Mention the existence of biomarkers used to assess oxidative stress.

- Paragraph on drug biotransformation (Sentences 103-105): Shorten the text with a brief mention of the potential link between drug metabolism and oxidative stress.

- How did the authors select the articles to be reviewed? Which database was used? The keywords used?

- Specific details of study design and methodology for each neuropsychiatric drug should be included. Was it a randomised controlled trial, an observational study or something else? How were the trials conducted? Sample size, dose used, control groups used, etc.

- The discussion does not distinguish between in vitro (cell) studies, animal studies and human clinical trials. This makes it difficult to assess the generalisability of the results to humans.

Author Response

Reviewer comment: This study aims to investigate the relationship between neuropsychiatric drugs and oxidative stress, a potential factor in several mental illnesses. The authors suggest that a better understanding of oxidative stress and its interaction with drugs may lead to more effective therapies.

Some suggestions:

- Paragraphs on oxidative stress (sentences 68-102): Shorten the text by briefly defining oxidative stress and its potential role in various health conditions, including mental illness. Mention the existence of biomarkers used to assess oxidative stress.

Response: Thank You for Your insight. The text was shorten defining only oxidative stress and its potential role in various health conditions, including mental illness. I also left few information about biomarkers used to assess oxidative stress.

- Paragraph on drug biotransformation (Sentences 103-105): Shorten the text with a brief mention of the potential link between drug metabolism and oxidative stress.

Response: Thank You for Your attention. We tried to shorten this fragment as much as possible, but not to lose the meaning.

- How did the authors select the articles to be reviewed? Which database was used? The keywords used?

Response: The data provided and discussed come from the US National Library of Medicine (PubMed) bibliographic sources, selecting publications from the last ten years and older for analysis in order to discuss the results of more recent studies. The key search terms for the literature search were oxidative stress and neuropsychiatric drugs.

- Specific details of study design and methodology for each neuropsychiatric drug should be included. Was it a randomised controlled trial, an observational study or something else? How were the trials conducted? Sample size, dose used, control groups used, etc.

Response: Thank you for this valuable comment, it is great. It actually gives us a better idea of ​​the content of each drug description.

- The discussion does not distinguish between in vitro (cell) studies, animal studies and human clinical trials. This makes it difficult to assess the generalisability of the results to humans.

Response: Thank you for this comment. We changed the conclusion and work out with the description. Although we tried to prepare such a large table collecting all the data, but unfortunately it turned out to be too extensive. To make the text easier to read, we propose a detailed study of individual drug descriptions, highlighting the pharmacological data of the drug and the results of preclinical and clinical studies. I hope this makes reading easier.
